# OsTBP2.1, a TATA-Binding Protein, Alters the Ratio of *OsNRT2.3b* to *OsNRT2.3a* and Improves Rice Grain Yield

**DOI:** 10.3390/ijms231810795

**Published:** 2022-09-16

**Authors:** Yong Zhang, Muhammad Faseeh Iqbal, Yulong Wang, Kaiyun Qian, Jinxia Xiang, Guohua Xu, Xiaorong Fan

**Affiliations:** 1State Key Laboratory of Crop Genetics and Germplasm Enhancement, MOA Key Laboratory of Plant Nutrition and Fertilization in Low-Middle Reaches of the Yangtze River, Nanjing Agricultural University, Nanjing 210095, China; 2Institute of Food Crops, Jiangsu Academy of Agricultural Sciences, Nanjing 210014, China

**Keywords:** *OsNRT2.3a*, *OsNRT2.3b*, TATA-box, OsTBP2.1, rice

## Abstract

The *OsNRT2.3a* and *OsNRT2.3b* isoforms play important roles in the uptake and transport of nitrate during rice growth. However, it is unclear which cis-acting element controls the transcription of *OsNRT2.3* into these specific isoforms. In this study, we used a yeast one-hybrid assay to obtain the TATA-box binding protein OsTBP2.1, which binds to the TATA-box of *OsNRT2.3*, and verified its important role through transient expression and RNA-seq. We found that the TATA-box of *OsNRT2.3* mutants and binding protein OsTBP2.1 together increased the transcription ratio of *OsNRT2.3b* to *OsNRT2.3a*. The overexpression of *OsTBP2.1* promoted nitrogen uptake and increased rice yield compared with the wild-type; however, the *OsTBP2.1* T-DNA mutant lines exhibited the opposite trend. Detailed analyses demonstrated that the TATA-box was the key cis-regulatory element for *OsNRT2.3* to be transcribed into *OsNRT2.3a* and *OsNRT2.3b*. Additionally, this key cis-regulatory element, together with the binding protein OsTBP2.1, promoted the development of rice and increased grain yield.

## 1. Introduction

Although the green revolution has made great progress in increasing the yield of grain crops, particularly rice, wheat, and maize, augmented cereal production has been accompanied by a huge increase in nitrogen fertilizer input that causes adverse effects to the environment [1,2]. With the continuous growth of the world’s population, increasing crop yield through environment-friendly and sustainable agriculture activities is vital to feed future populations [3]. Molecular breeding technology is a technological revolution that, in comparison to traditional breeding technology, can greatly improve the accuracy of crop breeding [4]. Therefore, a clear understanding of the mechanisms through which genes perform their functions is critical to optimize crops to effectively absorb nutrients.

The 5′ untranslated region (UTR) plays an important role in transcription and translation [5,6,7]. mRNA splicing and polyadenylation not only mutually affect efficiency and specificity but are also highly coordinated during transcription [8]. The interaction between alternative splicing (AS) and polyadenylation determines the outcome of gene expression [9]. In eukaryotes, AS is a key mechanism of post-transcriptional regulation that produces mRNAs with different stabilities, localizations, and translation efficiencies, thereby regulating gene expression [10,11,12] and greatly enhancing the diversity of the proteome [13,14]. AS can produce mRNAs with different UTRs or coding sequences, and these differences may lead to the synthesis of different isoforms of a protein or affect the stability/translatability of mRNA. The process of AS is deeply embedded in the regulatory networks that control development and differentiation processes in eukaryotes, and disorders in RNA splicing are the cause of several diseases in mammals [15,16]. Intron retention is the most frequent type of AS [17]. In the human nucleus, TARBP2 binding to pre-mRNAs leads to an increase in intron retention, which allows m^6^A to be recruited and deposited onto the target transcript, thereby affecting mRNA stability [18]. In plants, AS also affects developmental processes, and mutations affecting AS can be favorable for crop production. For example, in sweet maize, a C/GT mutation leading to intron retention in the 5′UTR of tiller number 1 (*Zmtin1*) enhances its transcript levels, consequently increasing *tin1* expression [19]. In Arabidopsis, intron retention in the 5′UTR of *ZIF2* promotes the response to zinc, enhancing tolerance to these metal ions [20].

In eukaryotes, there are one or more conserved sequences, such as TATA-box and T/A-rich sequences, bound by proteins in the promoter [21,22]. Moreover, the number of conserved motifs and mutations will affect the regulation or transcriptional activity of the gene and even affect the absorption and utilization of nutrients by plants. For example, the insertion of a TATA-box on the promoter of the iron-regulated transporter 1 (IRT1) gene in apples increases the expression of the gene IRT1. The greater the number of TATA-box insertions, the more the increase in multiple, thereby increasing iron uptake [23]. In cotton, when the TATA-box is absent from the gene PRE1D, cotton fiber elongation is reduced as the transcriptional activity of PRE1D decreases. In contrast, when the TATA-box is present, the upregulation of the transcriptional activity of the gene PRE1D increases cotton fiber elongation [24]. 

Nitrogen (N) is a key element in plant growth and development [25,26,27,28]. Plants absorb nutrients in different environments; this is closely related to the transcription and translation of their internal genes. However, in natural evolution, transcription factors play an important regulatory role in gene transcription and translation by binding to cis-acting elements on promoters [29,30]. Thus, studying the transcription factors and cis-acting elements can provide a better understanding of gene transcription and translation [31]. Previous studies have revealed that the mutation or deletion of a cis-acting element in a gene promoter can affect the binding ability of a transcription factor to a binding site, thereby affecting the transcriptional translation of the gene and even affecting the plant’s ability to absorb and utilize nutrients in different environments [29,32,33,34,35,36,37]. For example, OsNhd1 can directly bind to NBS-containing fragments (AAAAATCT and AGATTTTT), which, in the *OsNRT2.4* and *OsAMT1.3* promoters, activate their expression and affect N uptake efficiency in rice [32]. OsGhd7 (the rice grain number, plant height, and heading date 7 protein) directly binds to two evening element-like motifs in the promoter and intron 1 of *OsARE1* to repress its expression and regulate rice nitrogen utilization [33]. Therefore, transcription factors play an important role in the nitrogen uptake and utilization network in rice. 

In addition, in the nitrogen regulatory network, the *OsNRT2.3* gene plays a key role in rice growth, yield, and nitrogen utilization [38,39,40,41,42,43]. The *OsNRT2.3* gene, which encodes for a high-affinity nitrate transporter, produces two transcript isoforms, *OsNRT2.3a* and *OsNRT2.3b* [38,39]. We have reported the functional characterization of *OsNRT2.3a* and *OsNRT2.3b* in previous studies [38,39,40,41,42,43]. The products encoded by *OsNRT2.3a* and *OsNRT2.3b* are different, and the product encoded by *OsNRT2.3a* is 516 amino acids, which is 30 amino acids longer than that of *OsNRT2.3b* [38]. Additionally, the 5′UTRs of *OsNRT2.3a* and *OsNRT2.3b* are different, and the 5′UTR of *OsNRT2.3a* is 42 bp, while *OsNRT2.3b* has a 247 bp 5′UTR [38]. Rice genotypes with higher levels of *OsNRT2.3b* than *OsNRT2.3a* altered N absorption and transport efficiency factors that influence N-use efficiency (NUE) [38]. *OsNRT2.3a* reportedly participates in long-distance nitrate transport from root to shoot. However, *OsNRT2.3b*, which is primarily expressed in the phloem of the shoot, plays a role in pH and ion homeostasis and can cause membrane potential depolarization and cytoplasmic acidification under NO_3_^−^ supply conditions [38]. However, the mechanisms that regulate the transcription of *OsNRT2.3a* and *OsNRT2.3b* remain unknown. In this study, we show that the TATA-box binding protein OsTBP2.1 modulates the efficiency of the transcription initiation and splicing of *OsNRT2.3* to produce different levels of *OsNRT2.3a* and *OsNRT2.3b* isoforms. Our results uncover a mechanism for the regulation of *OsNRT2.3* that has a great impact on rice growth and yield, thereby offering novel directions for the molecular breeding of rice with improved yields.

## 2. Results

### 2.1. OsTBP2.1 Positively Regulates Rice Growth and Grain Yield in the Field

To determine whether *OsTBP2.1* influenced the growth and yield of rice, we produced transgenic rice lines that overexpressed *OsTBP2.1* (OE, OE198, and OE399) as well as *OsTBP2.1* knockout lines (*ostbp2.1*, 1A-19324, and 2B-30161) (Figure 1A,B; Appendix A). *OsTBP2.1* expression was evidently increased in the OE lines compared with the wild-type Wuyunjing 27 (WT-W27) and declined in the *ostbp2.1* lines compared with the wild-type HuangYang (WT-HY) (Figure 1B). In the field, we analyzed the agricultural traits of the OE and ostbp2.1 lines and found that the overexpression of *OsTBP2.1* improved rice growth and increased tiller number, plant height, and dry weight; however, the knockdown of *OsTBP2.1* elicited the opposite results (Appendix A; Figure 1C). Dry weight increased by approximately 43.2% in OE compared with that in WT-W27 but decreased in *ostbp2.1* by approximately 34.8% (Figure 1C). Rice yield is inextricably linked to panicle shape, and we analyzed the phenotype of the panicle at the maturity stage of rice. Grain number per panicle increased by approximately 30.2% in OE compared with that in WT-W27 but decreased in *ostbp2.1* by approximately 39.0% (Figure 1D). The panicle length of OE was increased by approximately 26.3% compared with that of WT-W27 but decreased in *ostbp2.1* by approximately 34.5% (Figure 1E). The seed setting rate of OE was increased by approximately 17.7% compared with that of WT-W27 but declined in *ostbp2.1* by approximately 66.2% (Figure 1F). The changes in panicle shape and seed setting rate can affect rice yield. This ultimately resulted in a 45.3% increase in the OE yield compared with that of WT-W27 and a 49.4% decrease in the yield of *ostbp2.1* compared with that of WT-HY (Figure 1G).

### 2.2. OsTBP2.1 Affects Nitrogen Uptake by Regulating the Transcription of OsNRT2.3a and OsNRT2.3b

To analyze the molecular biological functions of *OsTBP2.1*, we performed RNA-seq experiments using the OE lines and the *ostbp2.1* lines (Mu). The analysis of the differentially expressed genes (DEGs) revealed that there were 468 genes downregulated in the OE lines and upregulated in the *ostbp2.1* lines. However, 506 genes were downregulated in ostbp2.1 and upregulated in the OE lines in the root (Figure 2A). However, in the shoots, 509 genes were downregulated in the OE lines and upregulated in the *ostbp2.1* lines, and 318 genes were upregulated in the opposite case (Figure 2A). After filtering the gene expression data |log_2_FC| ≥2, Kyoto Encyclopedia of Genes and Genomes (KEGG) and Gene Ontology (GO) analyses revealed that 25 genes were regulated by *OsTBP2.1* in the rice roots and shoots (Figure 2B). Among the DEGs, *OsTBP2.1* affected genes that regulate N uptake or transport. *OsTBP2.1*, *OsNRT2.1*, *OsNRT2.2*, *OsNRT2.3a*, and *OsNRT2.3b* (Figure 2B; Appendix A) also had a strong effect on the transcription of *OsNRT2.3* in both the roots and shoots (Figure 2B and Figure 3).

To determine whether the expression of *OsTBP2.1* impacted the expressions of the *OsNRT2.3a* and *OsNRT2.3b* transcript isoforms, we analyzed the expression of *OsNRT2.3a* and *OsNRT2.3b* (Figure 3). We found that the OE lines exhibited increased overall expressions of *OsNRT2.3a* and *OsNRT2.3b*; however, the *ostbp2.1* lines exhibited the opposite trend (Figure 3A,B). Additionally, we observed that the OE lines increased the ratio of *OsNRT2.3b* to *OsNRT2.3a*, whereas the *ostbp2.1* lines decreased the ratio (Figure 3C). Overall, these results suggest that OsTBP2.1 regulates *OsNRT2.3* transcription, specifically enhancing the transcription of the *OsNRT2.3b* isoform, and that *OsTBP2.1* overexpression promotes rice growth.

As most of the N uptake by rice is distributed in the grain, rice with a higher N uptake capacity generally produces a higher yield. To explore how *OsTBP2.1* modulates N uptake in rice, we measured and analyzed the ^15^N-NH_4_^+^ and ^15^N-NO_3_^−^ influx rates in OE and *ostbp2.1*. The influx rate of ^15^N was decreased in *ostbp2.1* compared with that in WT-HY; however, it was increased under all ^15^N treatments in the OE lines compared with that in WT-W27 (Figure 4). Under the four treatments of 0.2 mM ^15^N-NH_4_^+^, 2.5 mM ^15^N-NH_4_^+^, 0.2 mM ^15^N-NO_3_^−^, and 2.5 mM ^15^N-NO_3_^−^, the influx rate of ^15^N in OE increased by approximately 110.0%, 62.5%, 306.0%, and 154.0%, respectively; however, in *ostbp2.1*, the ^15^N influx rate decreased by approximately 52.4%, 60.8%, 79.1%, and 75.6%, respectively (Figure 4). These results indicate that *OsTBP2.1* affects NH_4_^+^ and NO_3_^−^ uptake, particularly at low concentrations of N-NO_3_^−^.

### 2.3. OsTBP2.1 Bound to Cis-Element TATA-Box of OsNRT2.3

To verify OsTBP2.1, combined with the cis-acting elements of *OsNRT2.3*, we used the Softberry website (accessed on 1 May 2019) to analyze the sequence before the ATG of *OsNRT2.3*. We identified that the sequence of ATGCTATAAGAGC (from −89 to −77 bp upstream of *OsNRT2.3* ATG) was the motif for the TATA-box. We performed a yeast one-hybrid assay, using OsTBP2.1 as a DNA-binding protein and the TATA-box of *OsNRT2.3* as the binding site, to determine whether a TATA-box binding protein interacted with ATGCTATAAGAGC within the *OsNRT2.3* promoter. The results show that OsTBP2.1 can bind to the TATA-box of *OsNRT2.3*, indicated by growth in the SD/-Leu medium containing 800 ng/mL aureobasidin A (AbA^r^) (Figure 5A). To further determine whether OsTBP2.1 plays a role in the transcriptional activity of *OsNRT2.3* promoters and whether this effect is dependent on the TATA-box, we conducted transient expression assays using a dual-luciferase reporter assay system (Figure 5B). *OsTPB2.1* expression was controlled by the ubiquitin1 promoter, and the luciferase coding sequence was controlled by the *OsNRT2.3* promoter with C or T at position −83 bp upstream of *OsNRT2.3* ATG (Figure 5B). Both reporter and effector vectors were co-transformed into rice protoplasts. We found that OsTBP2.1 enhanced the expression of the pNRT2.3-LUC fusion genes in the WT promoter (pNRT2.3) but not in the mutant promoter (mpNRT2.3) (Figure 5C). These results showed that OsTBP2.1 promotes the transcriptional activity of the *OsNRT2.3* promoter and that the TATA-box of *OsNRT2.3* acts as a regulatory switch.

### 2.4. TATA-Box of OsNRT2.3 as a Regulatory Switch for OsNRT2.3 Transcription

To determine the mechanism through which OsTBP2.1 regulates the transcription of *OsNRT2.3a* and *OsNRT2.3b*, we structured expression vectors with different lengths of *OsNRT2.3* promoter fragments (Figure 6A). Transgenic seedlings of rice (*Nipponbare*) with promoter fragments of *OsNRT2.3* were obtained and transplanted into a hydroponic system supplied with 1.25 mM NH_4_NO_3_. The results showed that *OsNRT2.3a* expression was downregulated in the seedlings containing short promoter fragments, 141 and 180 bp. Moreover, *OsNRT2.3a* expression was upregulated in the seedlings containing long promoter fragments, 243, 697, and 1505 bp (Figure 6B). However, the relationship between promoter length and gene expression of *OsNRT2.3b* was the exact opposite of that observed for *OsNRT2.3a* (Figure 6C). By analyzing the ratio of *OsNRT2.3b* to *OsNRT2.3a*, we found that the short promoter fragments of *OsNRT2.3* promoted an increase in the ratio of *OsNRT2.3b* to *OsNRT2.3a*, whereas the long promoter fragments inhibited this effect (Figure 6D). Therefore, we can conclude that the transcription of *OsNRT2.3* to *OsNRT2.3b* is inhibited by the long fragment promoters of *OsNRT2.3*.

To explore the effect of TATA-box mutations on the transcription of *OsNRT2.3* into *OsNRT2.3a* and *OsNRT2.3b*, we obtained transgenic rice containing a TATA-box mutant (−83 bp mutation, ATGCTATAAGAGC mutant to ATGCTAcAAGAGC) with *OsNRT2.3* promoters of different lengths. The TATA-box mutant on the 141 bp fragment inhibited the transcription of both *OsNRT2.3a* and *OsNRT2.3b* compared with the natural 141 bp fragment (Figure 7A–C). However, the TATA-box mutant on the 697 bp fragment inhibited *OsNRT2.3a* transcription but promoted *OsNRT2.3b* transcription compared with the natural 697 bp fragment (Figure 7A–C). The proportion of *OsNRT2.3b* to *OsNRT2.3a* was increased when the TATA-box mutant was on fragments of 141 and 697 bp (Figure 7D). The results showed that the expression ratio of *OsNRT2.3b* to *OsNRT2.3a* was strongly correlated with the promoter length and specific TATA-box motif.

## 3. Discussion

The green revolution has improved crop yields, thereby meeting the demands of the growing world population [1,44,45,46]. Rice is one of the major cereals in demand, consumed by more than 1 billion people every day [47]. Therefore, understanding the molecular mechanisms underlying the transcription of genes that regulate rice productivity is important.

During plant growth, the response of plants to various environmental and developmental signals requires the precise expression of functional genes. Transcription factors play important roles in this process. For example, NLP transcription factors play a key role in nitrate regulation in higher plants [48]. In Arabidopsis, *NLP7*, a member of the RWPPK TF family, can bind to many nitrate signaling and assimilation genes to regulate their expression and response to plant primary nitrate [49,50,51,52]. The transcription factor *TCP20* regulates root foraging for nitrate by binding to the nitrate enhancer fragment of *NIA1* and *NRT2.1* [53,54,55]. In rice, the TATA-box binding protein OsTBP2.1, a ubiquitous transcription factor, combines with the cis-acting element −89 to −77 bp upstream of the gene *OsNRT2.3* ATG and regulates the transcription of *OsNRT2.3* to *OsNRT2.3a* and *OsNRT2.3b* (Figure 3; Figure 5). Transcription factors regulate gene transcription by binding to cis-acting elements. Therefore, these cis-acting elements play an important role in the transcriptional regulation of each gene, and mutations in key cis-acting elements affect gene regulation. For example, *OsMADS57*, a MADS-box transcription factor, can bind to the CArG motif [CATTTTATAG] on the *OsNRT2.3a* promoter and regulate nitrate translocation from roots to shoots [56]. The cis-acting element TATA-box controls the transcription of *OsNRT2.3* via binding with OsTBP2.1. In this study, when the TATA-box was mutated, *OsTBP2.1* decreased the binding ability and altered the transcription pattern of *OsNRT2.3*, thereby increasing the proportion of *OsNRT2.3b* to *OsNRT2.3a*
Figure 5B,C and Figure 7). The TATA-box in *OsNRT2.3b* plays an important regulatory role in the splicing pattern of *OsNRT2.3* to *OsNRT2.3a* and *OsNRT2.3b*, and the *OsNRT2.3* 5′UTR mutation changed the splicing mode of *OsNRT2.3*.

In addition, the 5′UTR plays regulatory roles in RNA translation, stability, and transcription. The number and length of introns in the 5′UTR can influence gene expression [57]. Key cis-acting elements that are also on the 5′UTR containing the first intron exist on the promoter, and introns can enhance transcription in plants [58,59]. The 5′UTR containing the first intron upstream of *ZmBCH2* can enhance *gusA* expression in transgenic rice [60]. In our study, we found that a complete promoter of *OsNRT2.3* promoted the expression of both *OsNRT2.3a* and *OsNRT2.3b*; however, the presence of the intron at the 5′UTR of *OsNRT2.3a* promoted *OsNRT2.3a* expression and inhibited *OsNRT2.3b* transcription. However, when there was a mutation in the TATA-box of *OsNRT2.3*, the natural balance was disrupted (Figure 6 and Figure 7). In natural variation and selection, species tend to develop in favorable directions. Mutations in *OsNRT2.3b* 5′UTR can accelerate breeding and increase NUE. The different isoforms of *OsNRT2.3* are expressed in different locations, with *OsNRT2.3b* primarily expressed at the shoot and *OsNRT2.3a* primarily expressed at the root [38]. This mechanism also reveals how *OsNRT2.3* is efficiently transcribed into its two isoforms. This mechanism may be affected by other factors, such as the length of the *OsNRT2.3* promoter and the binding ability of OsTBP2.1 (Figure 5B,C, Figure 6 and Figure 7). Rice genotypes with higher levels of *OsNRT2.3b* than *OsNRT2.3a* can alter N absorption and transport efficiency factors that influence NUE and yield [38]. High expression of *OsTBP2.1* promoted rice growth and increased yield, whereas low expression exhibited the opposite trend ( Figure 1, Appendix A). Therefore, OsTBP2.1 positively regulates rice growth, yield, and N uptake by changing the ratio of *OsNRT2.3* transcription to *OsNRT2.3b* and *OsNRT2.3a* (Figure 1, Figure 3 and Figure 4).

Additionally, it is generally believed that transcription factor IID (TFIID), including TBP-associated factor (TAFs) and TATA-binding protein (TBP) genes, is mandatory for the precise transcription initiation of core promoters, which limited their role only to transcriptional activity. However, this was disproved; it was revealed that TAFs are required for plant growth. Moreover, each TAF is needed for the transcription of a limited subset of genes, ranging from 3% to 67% of the total number of expressed genes [61]. Alterations in conserved transcription factors typically alter transcription patterns or transcription levels of multiple genes. For example, the knockdown of *OsTBP2.2*, a gene belonging to the same family as OsTBP2.1, affects multiple main pathways and downregulates the expression of *OsPIP2;6*, *OsPAO*, and *OsRCCR1* genes, thereby inhibiting photosynthesis and growth in rice [62]. Altering the transcript abundance of *OsTBP2.1* altered the expression of nitrogen-related genes, such as *OsNRT2.1* (*Os02g0112100*), *OsNRT2.2* (*Os02g0112600*), *OsNRT2.3a*, and *OsNRT2.3b*, thereby affecting nitrogen uptake and yield in rice (Figure 2, Figure 3, Figure 4 and Appendix A). OsTBP2.1 is the regulatory switch for the transcription of *OsNRT2.3*, *OsNRT2.3a*, and *OsNRT2.3b*.

Another way in which *OsTBP2.1* regulates rice yield may be through the regulation of the F-box/FBD/LRR-repeat protein gene *Os070158900*. The spatial and temporal gene expression of F-box proteins is induced during seed development and panicle formation [63], regulated by the mRNA expression of *OsTBP2.1* (Figure 1 and Figure 2). These possible pathways between growth-regulating proteins from the F-Box and LRR families are upregulated by *OsTBP2.1* in accordance with GO and KEGG and predict the role of *OsTBP2.1* in reproductive growth and seed formation.

## 4. Conclusions

In summary, this study reveals that the length of the *OsNRT2.3* promoter and the TATA-box cis-acting element together affect the ratio of *OsNRT2.3* transcription to *OsNRT2.3b* and *OsNRT2.3a*. Additionally, the TATA-box binding protein OsTBP2.1 can affect the growth and yield of rice by combining with the *OsNRT2.3* TATA-box to regulate the transcription of *OsNRT2.3a* and *OsNRT2.3b*. This comprehensive understanding of how *OsNRT2.3a* and *OsNRT2.3b* are regulated reveals a novel path for molecular research and the molecular breeding of plants.

## 5. Materials and Methods

### 5.1. Plant Materials

In this study, the Huang Yang (WT-HY), Wuyunjing 27 (WT-W27), and Nipponbare (WT-N) rice ecotypes were used. The T-DNA insertion mutant lines, 1A-19324 and 2B-30161, belong to the WT-HY ecotype. T-DNA insertion mutant lines were obtained from Hyung Hee University, Korea. We used the WT-W27 ecotype to create the *OsTBP2.1* overexpression lines, OE198 and OE399. We used the *pUbi* promoter to construct the *OsTBP2.1* overexpression vector (*pUbi*::*OsTBP2.1*) in the pTCK303 expression vector and transcribed this into the WT-W27 ecotype to obtain the *OsTBP2.1* overexpression lines.

The *OsNRT2.3* promoters were obtained from the WT-N ecotype. We amplified different lengths (such as 697 and 141 bp) of the *OsNRT2.3* promoter from Nipponbare. We generated a −83 bp mutant in the 141 and 697 bp promoters to investigate whether this affected the expression ratio of *OsNRT2.3a* and *OsNRT2.3b*. The promoters of different lengths drove the 437 bp open reading frame (ORF) of *OsNRT2.3*. To better differentiate between *OsNRT2.3a* and *OsNRT2.3b*, we linked the 437 bp ORF with the sequence of ZǀǀǀB (cctgcaggtcgccac attagcaatgccacattagcaatgccgactctagaggatccc) in pCAMBIA1300. The expression vectors were transferred into Agrobacterium tumefaciens (EHA105) via electroporation and then transformed into rice. The primers are shown in Appendix A.

### 5.2. Plant Growth Conditions

For the field experiments, the lines used in this study were grown in plots at Nanjing Agricultural University in Nanjing, Jiangsu. The chemical properties of the soils in the plots have been described by Chen et al. [64].

For hydroponic experiments, the lines used in this study were surface-sterilized with 30% (*v/v*) NaClO for 30 min and then rinsed thoroughly with water. Seedlings were grown in an artificial climate chamber with a 16 h light (30 °C) and 8 h dark (26 °C) cycle.

### 5.3. Southern Blot Analysis

Southern blotting was performed as described [65]. Briefly, genomic DNAs of *OsTBP2.1* overexpression lines and WT-W27 were extracted and digested with the restriction enzymes *Hind* III and *EcoR* I overnight. The digested genomic DNAs were then transferred to a Hybond-N^+^ nylon membrane. A hygromycin-resistant probe was used to hybridize the digested genomic DNAs onto a Hybond-N^+^ nylon membrane.

### 5.4. ^15^N Uptake Experiments

^15^N influx was conducted to determine ^15^NH_4_^+^ and ^15^NO_3_^−^ uptake in the lines of the T-DNA insertion mutant, *OsTBP2.1* overexpression lines, and wild-type plants (WT-HY and WT-W27). The plants were grown in hydroponics with IRRI nutrient solution for two weeks, followed by N starvation for three days. After N starvation, the plants were initially moved to 0.1 mM CaSO_4_ for 1 min, and they were then separately exposed to 0.2 mM ^15^NH_4_^+^, 0.2 mM ^15^NO_3_^−^, 2.5 mM ^15^NH_4_^+^, and 2.5 mM ^15^NO_3_^−^ for 5 min each. The seedlings were then transferred to 0.1 mM CaSO_4_ for 1 min before sampling. All plants were maintained at 105 °C for 30 min to inactivate the enzymes. The powder of each sample was analyzed using an isotope ratio mass spectrometer system (Flash 2000 HT, Thermo Fisher Scientific, Brunswick, Germany). 

### 5.5. RNA-Seq Analysis

Total RNA was extracted from the shoots and roots of the wild-type plants (WT-27, WT-HY), *OsTBP2.1* overexpression lines (OE), and *OsTBP2.1* T-DNA insertion mutant lines (Mu), with three biological replicates. The samples were tested using Genepioneer software. RNA-seq analysis was performed using an Illumina HiSeq 2000 Plus instrument. The filtered values and reads were aligned with the reference genome using HISAT2 software. HTSeq was used to align the read count values for every gene in comparison to the original expression level of the gene, and the expression level was then normalized using FPKM or fragments differentially expressed between wild-type and transgenic lines. DEGs were defined as genes with an expression difference of (log_2_FoldChange) >2 and a *p*-value < 0.05. The identified DEGs were further analyzed using KEGG, GO, nr-annotations, and Swissport annotations [62,66].

### 5.6. Quantitative Real-Time PCR

Total RNA was extracted from the field and hydroponic experiments using TRIzol reagent. The extracted RNA was converted into cDNA using the Hiscript Q RT SuperMix for qPCR (+dDNA wiper) kit (Vazyme, Co. R323-01, Nanjing, China). The synthesized cDNA was used as a template in real-time PCR reactions using the AceQ qPCR SYBR Green Master Mix kit (Vazyme, Co. Q311-02, Nanjing, China) and the Step One Plus real-time PCR system (Applied Biosystems, Foster City, CA, USA). The primers are shown in Appendix A.

### 5.7. Yeast One-Hybrid Assay

The yeast one-hybrid assay was performed according to the manufacturer’s protocol in the “Matchmaker Gold Yeast One-Hybrid Library Screening System User Manual” (Clontech). Briefly, the pTATA-box-pAbAi vector was transferred into the yeast strain and grown on medium lacking uracil (Ura). We used different concentrations of aureobasidin A (AbA^r^) to test the bait strain in a medium lacking Ura. The vectors of pGADT7:OsTBP2.1 were then transferred to the strains with pTATA-box-pAbAi, and the strains were grown on a medium lacking leucine with 800 ng mL^−1^ AbA^r^.

### 5.8. Promoter Activity Analysis

The 1.5 kb promoter and -83 bp mutation on the 1.5 kb promoter fragments of *OsNRT2.3* were amplified from Nipponbare and inserted into the luciferase reporter. The plasmids were transferred into rice protoplasts together with pUbi::OsTBP2.1 and harvested 24 h later. The protoplasts were analyzed using the dual-luciferase reporter assay system (Promega, Cat. E2920) to calculate the ratio of firefly luciferase (LUC) to Renilla (REN) luciferase. 

### 5.9. Statistical Analysis

Data are presented as mean ± standard error (SE) and were analyzed by ANOVA using the statistical software SPSS (version 11.0; SPSS Inc., Chicago, IL, USA). Significant differences were determined using the SPSS Statistics 20 program and one-way ANOVA, followed by Tukey’s test (*p* < 0.05, one-way ANOVA).

## Figures and Tables

**Figure 1 ijms-23-10795-f001:**
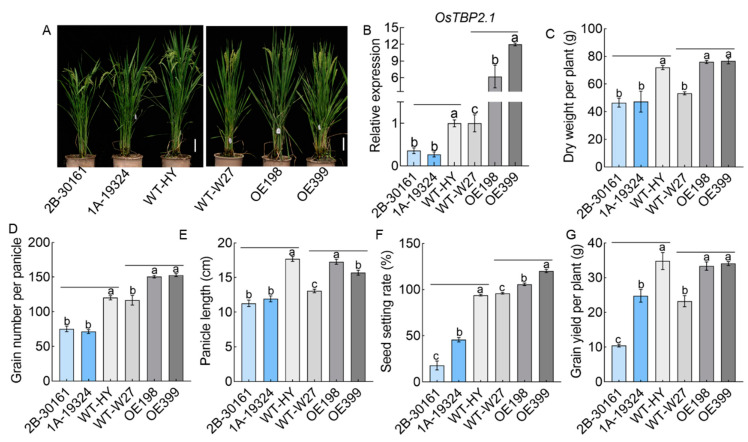
Phenotype of *OsTBP2.1* T-DNA insertion mutant and *OsTBP2.1* overexpression lines. (**A**) Characterization of T-DNA insertion mutant lines (*ostbp2.1*, 1A-19324 and 2B-30161) and *OsTBP2.1* overexpression lines (OE, OE198 and OE399) in a field experiment. Wild-type, Huangyang (WT-HY) and Wuyunjing27 (WT-W27). (Bar = 20 cm). (**B**) The expression of *OsTBP2.1* in the OE lines and *ostbp2.1* lines. Error bars: SE (n = 3). At the maturity stage of OE lines and *ostbp2.1* lines, dry weight (**C**), grain number per panicle (**D**), panicle length (**E**), seed setting rate (**F**), and grain yield per plant (**G**) were taken from the field. Error bars: SE (n = 5). The different letters indicate a significant difference between each other (*p* < 0.05).

**Figure 2 ijms-23-10795-f002:**
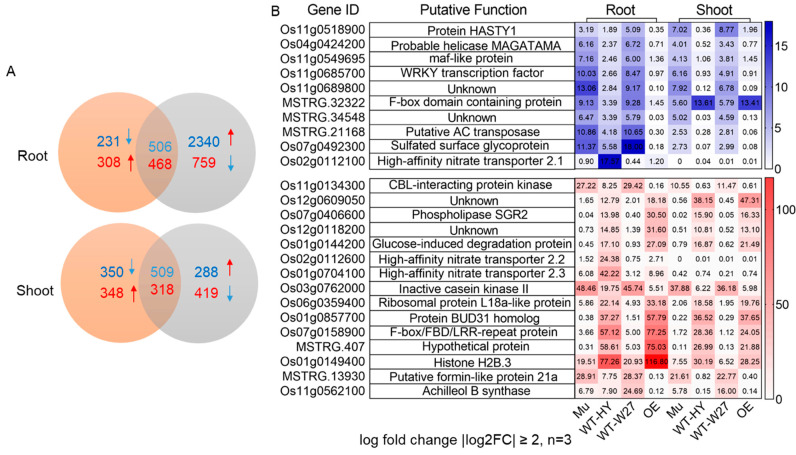
RNA-seq analysis of *OsTBP2.1* overexpression lines and T-DNA insertion mutant lines. (**A**) The number of different expression genes (DEGs) identified by the RNA-seq dataset in the roots and shoots of OE and *ostbp2.1*. The samples taken from the shoots and roots of *OsTBP2.1* overexpression lines (OE) and T-DNA insertion mutant lines (*ostbp2.1*). Blue arrow, downregulated genes in OE and *ostbp2.1*. Red arrow, upregulated genes in OE and *ostbp2.1*. (**B**) Individual accession number, putative function, and relative expression of the 25 genes selected from (**A**). Log fold change |log_2_FC|≥2 of the RNA-seq analysis (n = 3).

**Figure 3 ijms-23-10795-f003:**
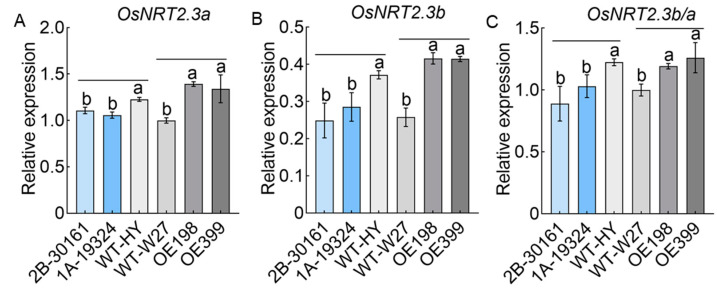
The expression of *OsNRT2.3a* and *OsNRT2.3b* in the T-DNA insertion mutant lines and *OsTBP2.1* overexpression lines. The expression of *OsNRT2.3a* (**A**) and *OsNRT2.3b* (**B**) in the OE lines and *ostbp2.1* lines. (**C**) The expression ratio of *OsNRT2.3b* to *OsNRT2.3a* in the OE lines and *ostbp2.1* lines. Error bars: SE (n = 3). Significant differences between *ostbp2.1* and WT-HY, OE, and WT-W27 are indicated by different letters (*p* < 0.05).

**Figure 4 ijms-23-10795-f004:**
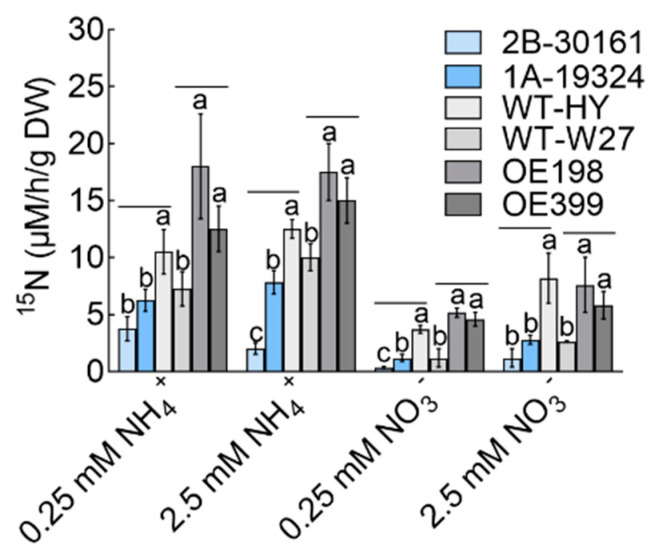
Effects of *OsTBP2.1* overexpression lines and T-DNA insertion mutant lines on the root influx of ^15^N for 5 min. *OsTBP2.1* overexpression lines (OE) and T-DNA insertion mutant lines (*ostbp2.1*) were grown in 1.25 mM NH_4_NO_3_ for 3 weeks and nitrogen starved for 3 days. The ^15^N influx rate was then measured at 0.25 mM ^15^NH_4_^+^, 2.5 mM ^15^NH_4_^+^, 0.25 mM ^15^NO_3_^−^, and 2.5 mM ^15^NO_3_^−^ over a 5 min period. Error bars: SE (n = 5). The different letters indicate a significant difference between the transgenic line and the wild-type (*p* < 0.05).

**Figure 5 ijms-23-10795-f005:**
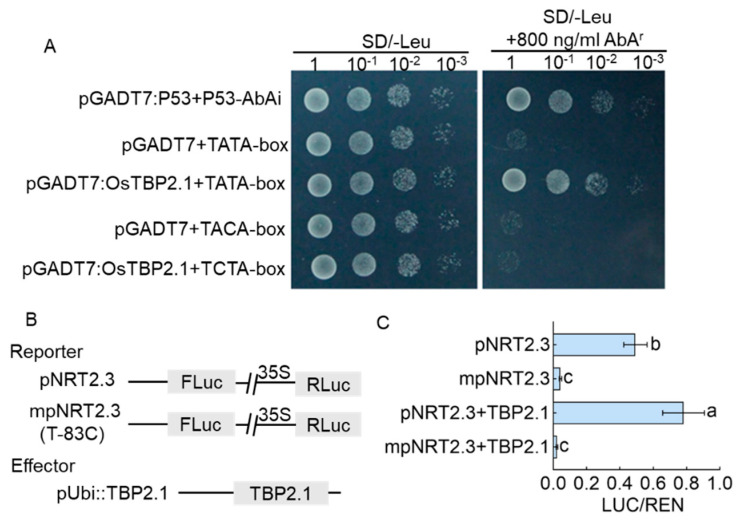
OsTBP2.1 binds to the cis-acting element of *OsNRT2.3* (**A**) OsTBP2.1 binding to the TATA-box of *OsNRT2.3*. Yeast cells were co-transformed with *p*TATA-box::AbAi and *p*GADT7::OsTBP2.1. *p*53::AbAi and *p*GADT7::P53 were the positive controls. *p*TACA-box::AbAi and *p*GADT7::OsTBP2.1 were the negative controls. Cells were grown on the media with selection for interaction (SD, -Leu) and (800 ng/mL) AbA^r^ to suppress background growth. (**B**) Constructs of the rice protoplast transient assay. The *OsNRT2.3* promoter (*p*NRT2.3::LUC) or TATA-box mutated (TATA-box mutant to TACA-box) promoter (*mp*NRT2.3::LUC) drove the reporter. pUbi::TBP2.1 was the effector. The effector of OsTBP2.1 was driven by the Ubi promoter. The reporter and effector were co-transformed into the rice protoplast. (**C**) OsTBP2.1 affected the *OsNRT2.3* promoter’s activation. Error bars: SE (n = 3). The different letters indicate a significant difference between each other (*p* < 0.05).

**Figure 6 ijms-23-10795-f006:**
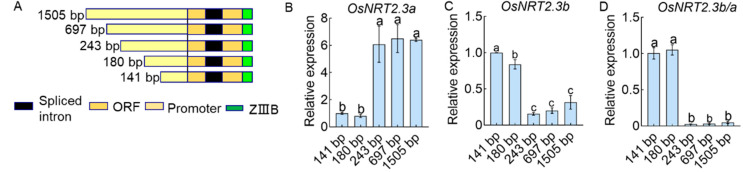
Effect of different *OsNRT2.3* promoter lengths on *OsNRT2.3a* and *OsNRT2.3b* expression in rice. (**A**) Schematic representation of the different *OsNRT2.3* promoter fragments 141, 180, 243, 697, and 1505 bp, driving the expression of the 437 bp open reading frames (ORF) of *OsNRT2.3* and the *ZIIIB* reporter gene. Black frames, spliced intron. Brown frames, ORF of *OsNRT2.3a* and *OsNRT2.3b*. Yellow frames, promoters of *OsNRT2.3*. Green frames, *ZIIIB* reporter gene, division *OsNRT2.3a* and *OsNRT2.3b*. Obtained transgenic lines planted in a solution containing 1.25 mM NH_4_NO_3_ of IRRI. The expression of *OsNRT2.3a* (**B**) and *OsNRT2.3b* (**C**) in the lines of different lengths of *OsNRT2.3* promoters was derived. (**D**) The expression ratio of *OsNRT2.3b* to *OsNRT2.3a* in the lines of different lengths of *OsNRT2.3* promoters was derived. Error bars: SE (n = 3). Significant differences between the lines are indicated by different letters (*p* < 0.05).

**Figure 7 ijms-23-10795-f007:**
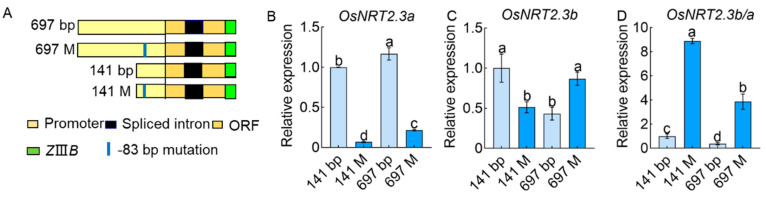
Effects of TATA-box mutation at different length promoters influence the transcription of *OsNRT2.3* in rice. (**A**) Schematic representation of the diagram of different lengths of *OsNRT2.3* promoter fragments; TATA-box mutation fragments promote the 437 bp ORF of *OsNRT2.3* and the *ZIIIB* reporter gene; 141 and 697 bp are the different lengths of the original *OsNRT2.3* promoter; 141M and 697M are the different lengths of the *OsNRT2.3* promoter containing the TATA-box mutation site. Yellow frames, promoters of *OsNRT2.3*. Black frames, spliced intron. Brown frames, ORF of *OsNRT2.3a* and *OsNRT2.3b*. Green frames, *ZIIIB* reporter gene. Blue line, TATA-box mutation site, −83 bp upstream of *OsNRT2.3* ATG. Obtained transgenic lines planted in a solution containing 1.25 mM NH_4_NO_3_ of IRRI. The expression of *OsNRT2.3a* (**B**) and *OsNRT2.3b* (**C**) in the transgenic lines. (**D**) The expression ratio of *OsNRT2.3b* to *OsNRT2.3a* in the transgenic lines. Error bars: SE (n = 3). The different letters indicate a significant difference between the transgenic lines (*p* < 0.05).

## Data Availability

All data in this article is available from the corresponding author upon reasonable request.

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
