# Peer review of "OsTBP2.1, a TATA-Binding Protein, Alters the Ratio of OsNRT2.3b to OsNRT2.3a and Improves Rice Grain Yield"

_ijms, 2022, doi:10.3390/ijms231810795_

Round 1
Reviewer 1 Report
This study investigated the function of the TATA-binding protein OsTBP2.1 in the regulation of the nitrate transporters OsNRT2.3b/a and rice grain yield. It was found that OsTBP2.1 bound to the TATA-box of OsNRT2.3. Overexpressing OsTBP2.1 promoted nitrogen uptake and increased rice yield. I have the following suggestions to improve this manuscript.
1. Please explain in more detail what is the difference between OsNRT2.3a and OsNRT2.3b. Do OsNRT2.3a and OsNRT2.3b.transcripts produce different proteins?
2. Figure 6A and Figure 7A
Also, please explain what is the ZIIIB reporter gene and why it was used here.
3. Figure 6 and Figure 7
Please explain how the relative expression of OsNRT2.3a and OsNRT2.3b was carried out. How many transgenic plants were analyzed?
4. English language needs to be improved. For examples:
Line 150 – Please change “we taken the expression of OsNRT2.3a and OsNRT2.3b” to “we analyzed the expression of OsNRT2.3a and OsNRT2.3b”.
Line 221 – Please change “up-regulated in the seedlings which OsNRT2.3 promoted by longer promoter fragments” to “up-regulated in the seedlings in which OsNRT2.3 was promoted by longer promoter fragments”.
Author Response
Thank you for your valuable comments.
This study investigated the function of the TATA-binding protein OsTBP2.1 in the regulation of the nitrate transporters OsNRT2.3b/a and rice grain yield. It was found that OsTBP2.1 bound to the TATA-box of OsNRT2.3. Overexpressing OsTBP2.1 promoted nitrogen uptake and increased rice yield. I have the following suggestions to improve this manuscript.
- Q. Please explain in more detail what is the difference between OsNRT2.3a and OsNRT2.3b. Do OsNRT2.3a and OsNRT2.3b transcripts produce different proteins?
A: Thank you for your valuable comments. OsNRT2.3a and OsNRT2.3b are different in gene structure and functional area.
In lines 86-90 add ‘The products encoded by OsNRT2.3a and OsNRT2.3b are different, and the product encoded by OsNRT2.3a is 516 amino acids, which is 30 amino acids longer than that of OsNRT2.3b [38]. And the 5’UTR of OsNRT2.3a and OsNRT2.3b are different, and the 5’UTR of OsNRT2.3a is 42 bp, while OsNRT2.3b with a 247 bp 5’UTR [38].’ OsNRT2.3a encodes a plasma membrane protein that functions in long-distance nitrate transport in the xylem from root to shoot, whereas OsNRT2.3b encodes a plasma membrane protein expressed moderately in the phloem of the shoot and faintly in the root.
- Q. Figure 6A and Figure 7A
Also, please explain what is the ZIIIB reporter gene and why it was used here.
A: The sequence of ZIIIB reporter is a specific sequence which the protein ZEBRA specific binding. The OsNRT2.3a and OsNRT2.3b sequences are as high as 88% similar. In this study, to better differentiate between OsNRT2.3a and OsNRT2.3b, we used the reporter ZIIIB (cctgcaggtcgccacattagcaatgccacattagcaatgccgactctagaggatccc).
- Q. Figure 6 and Figure 7
Please explain how the relative expression of OsNRT2.3a and OsNRT2.3b was carried out. How many transgenic plants were analyzed?
A: Thank you for your valuable comments. Forward primer which used to detect the abundance of OsNRT2.3a or OsNRT2.3b expression in qRT-PCR is designed on the ORF sequence of OsNRT2.3a or OsNRT2.3b. And the reverse primer is designed on the ZIIIB reporter sequence. The primer details are as follows:
OsNRT2.3a Forward primer: GCTCATCCGCGACACCCT
OsNRT2.3b Forward primer: CACGTTCGCCGTGTTC
Reverse primer: ATTGCTAATGTGGCGACCT
We used three transgenic plants with three biological replicates each for analysis.
- Q. English language needs to be improved. For examples:
Line 150 – Please change “we taken the expression of OsNRT2.3a and OsNRT2.3b” to “we analyzed the expression of OsNRT2.3a and OsNRT2.3b”.
Line 221 – Please change “up-regulated in the seedlings which OsNRT2.3 promoted by longer promoter fragments” to “up-regulated in the seedlings in which OsNRT2.3 was promoted by longer promoter fragments”.
A: Thank you for your valuable comments. The language of the manuscript have been improved by a native speaker to make sure good enough for publication.

Reviewer 2 Report
The authors have completed a sound investigation of the mechanisms of the TATA-box binding protein, OsTBP2.1 that binds to the TATA-box of OsNRT2.3, which may have implications for nitrogen uptake and increased rice yield.
The only other studies I can forsee that the researchers may have conducted, would be targeted CRISPR knockouts of specific regions of the OsTBP2.1, to see if the the expression levels of OsNRT2.3 isomers were altered. But such work was attempted with the creation of several expression vectors with differing lengths of promoter fragments.
In its current state, I do not think the manuscript is ready for publishing, I have flagged that moderate english changes is required. The paper requires the authors to go through the manuscript and get the tense of the english correct e.g. bind vs bound etc. Likewise, there are several examples where sentences are too long or need re-writing e.g. Line 243 to 249 and line 335 to 337. Some basic errors are also present e.g. "Rice is one of the major cereals, meeting the demands of more than 10 billion people every day", there is only 7.753 billion people on the planet.
The other issue I have with the manuscript is the Methods section. There is no figures illustrating the vectors used in the study, list of primers used for qPCR, detail on the different promoter sequences selected for the expression vectors apart from their length etc. All this means that another researcher couldn't replicate some of the experiments laid out in this paper.
Author Response
The authors have completed a sound investigation of the mechanisms of the TATA-box binding protein, OsTBP2.1 that binds to the TATA-box of OsNRT2.3, which may have implications for nitrogen uptake and increased rice yield.
- Q. 1 The only other studies I can forsee that the researchers may have conducted, would be targeted CRISPR knockouts of specific regions of the 1, to see if the the expression levels of OsNRT2.3 isomers were altered. But such work was attempted with the creation of several expression vectors with differing lengths of promoter fragments.
A: Thank you for your valuable comments. In this study, we built the expression vectors with TATA-box mutants and different lengths of OsNRT2.3 promoter fragments, and transgenic to Japonica rice. In generation T4, we detected the expression of OsNRT2.3a and OsNRT2.3b with the primers shown in Fig. S5 which can be distinguish between OsNRT2.3a and OsNRT2.3b. The OsNRT2.3a and OsNRT2.3b sequences are as high as 88% similar. To better differentiate between OsNRT2.3a and OsNRT2.3b, we used the reporter ZIIIB (cctgcaggtcgccacattagcaatgccacattagcaat gccgactctagaggatccc). Forward primer which used to detect the abundance of OsNRT2.3a or OsNRT2.3b expression in qRT-PCR is designed on the ORF sequence of OsNRT2.3a or OsNRT2.3b. And the reverse primer is designed on the ZIIIB reporter sequence.
- Q. 2 In its current state, I do not think the manuscript is ready for publishing, I have flagged that moderate english changes is required. The paper requires the authors to go through the manuscript and get the tense of the english correct e.g. bind vs bound etc. Likewise, there are several examples where sentences are too long or need re-writing e.g. Line 243 to 249 and line 335 to 337. Some basic errors are also present e.g. "Rice is one of the major cereals, meeting the demands of more than 10 billion people every day", there is only 7.753 billion people on the planet.
A: Thank you for your valuable comments. The language of the manuscript have been improved by a native speaker to make sure good enough for publication.
- Q. 3 The other issue I have with the manuscript is the Methods section. There is no figures illustrating the vectors used in the study, list of primers used for qPCR, detail on the different promoter sequences selected for the expression vectors apart from their length etc. All this means that another researcher couldn't replicate some of the experiments laid out in this paper.
A: Thank you for your valuable comments. In the line 353, we add the vector information, ‘We used the pUbi promoter to construct the OsTBP2.1 overexpression vector (pUbi::OsTBP2.1) in pTCK303 expression vector’.
In the line 359, we add the vector information, ‘To better differentiate between OsNRT2.3a and OsNRT2.3b, we linked the 437 bp ORF with the sequence of ZǀǀǀB (cctgcaggtcgccac attagcaatgccacattagcaatgccgactctagaggatccc) in pS1aGUS-3.’
We present sequence information for primers in Table S1 and Table S2.
